# Current Insight into Culture-Dependent and Culture-Independent Methods in Discovering Ascomycetous Taxa

**DOI:** 10.3390/jof7090703

**Published:** 2021-08-28

**Authors:** Nalin N. Wijayawardene, Mohammad Bahram, Iván Sánchez-Castro, Dong-Qin Dai, Kahandawa G. S. U. Ariyawansa, Udeni Jayalal, Nakarin Suwannarach, Leho Tedersoo

**Affiliations:** 1Centre for Yunnan Plateau Biological Resources Protection and Utilization, College of Biological Resource and Food Engineering, Qujing Normal University, Qujing 655011, China; nalinwijayawardene@yahoo.com; 2State Key Laboratory of Functions and Applications of Medicinal Plants, Guizhou Medical University, Guiyang 550014, China; 3Section of Genetics, Institute for Research and Development in Health and Social Care, No. 393/3, Lily Avenue, Off Robert Gunawardane Mawatha, Battaramulla 10120, Sri Lanka; 4Department of Ecology, Swedish University of Agricultural Sciences, Ulls väg 16, 756 51 Uppsala, Sweden; bahram@ut.ee; 5Institute of Ecology and Earth Sciences, University of Tartu, 14A Ravila, 50411 Tartu, Estonia; 6Departamento de Microbiología, Campus de Fuentenueva, Universidad de Granada, 18071 Granada, Spain; ivansanchezcastro@gmail.com; 7Department of Plant Sciences, Faculty of Science, University of Colombo, Colombo 00300, Sri Lanka; sameera@pts.cmb.ac.lk; 8Department of Natural Resources, Sabaragamuwa University of Sri Lanka, Belihuloya 70140, Sri Lanka; jayalal@appsc.sab.ac.lk; 9Research Center of Microbial Diversity and Sustainable Utilization, Faculty of Science, Chiang Mai University, Chiang Mai 50200, Thailand; suwan.462@gmail.com; 10Mycology and Microbiology Center, University of Tartu, 14A Ravila, 50411 Tartu, Estonia; leho.tedersoo@ut.ee; 11College of Science, King Saud University, Riyadh 11451, Saudi Arabia

**Keywords:** consolidated species concept, fungal diversity, molecular taxonomy, morphological characters, traditional taxonomy, polyphasic approach

## Abstract

Culture techniques are vital in both traditional and modern fungal taxonomy. Establishing sexual–asexual links and synanamorphs, extracting DNA and secondary metabolites are mainly based on cultures. However, it is widely accepted that a large number of species are not sporulating in nature while others cannot be cultured. Recent ecological studies based on culture-independent methods revealed these unculturable taxa, i.e., dark taxa. Recent fungal diversity estimation studies suggested that environmental sequencing plays a vital role in discovering missing species. However, Sanger sequencing is still the main approach in determining DNA sequences in culturable species. In this paper, we summarize culture-based and culture-independent methods in the study of ascomycetous taxa. High-throughput sequencing of leaf endophytes, leaf litter fungi and fungi in aquatic environments is important to determine dark taxa. Nevertheless, currently, naming dark taxa is not recognized by the ICN, thus provisional naming of them is essential as suggested by several studies.

## 1. Introduction

Fungi form diverse communities in nature and also play essential roles in many ecosystems. Species Fungorum [1] currently includes 150,048 species with valid names, but Hawksworth and Lücking [2] estimated the total number of species to be between 2.2 and 3.8 million. Only 6.8% to 3.9% of all species are currently known, and numerous species remain to be described. Isolating and maintaining cultures of these taxa (which are from different habitats) have been regarded as vital steps in modern taxonomy as these cultures are widely used to obtain DNA. Nevertheless, numerous taxa from different substrates (e.g., soil, sea water, and sediments) have been reported as unculturable, and thus different methods have been developed to screen them (e.g., ichip *fide* Nichols et al. [3] and Berdy et al. [4]). As a technique, direct extraction from a complex substrate—the so-called eDNA approach—of DNA is increasingly used in studying various groups of fungi such as lichens, obligate parasitic taxa, and other unculturable taxa. In particular, studies of fungal diversity have incorporated more and more DNA-based molecular techniques and polyphasic approaches in the past three decades (1990–2020), resulting in considerable advances in the field. In these methods, species identification, linking sexual–asexual morphs (of pleomorphic taxa), and higher-level classification are performed based on DNA sequences. These unculturable taxa are important in studying missing taxa in the Kingdom *Fungi* [5]. Moreover, fungi which produce fruiting structures (e.g., ascomata and conidiomata in ascomycetous fungi and basidiomata in basidiomycetous fungi) might produce spores that are incapable of germinating and thus would not yield cultures. For example, Dai et al. [6] introduced *Rubroshiraia*, a novel genus. The type species, *Rubroshiraia bambusae* was incapable of germinating, thus DNA was extracted directly from the fruiting structures.

During the last decade, approximately 1200–2000 fungal species have been described annually, the majority of them being ascomycetes [pers. com. P.M. Kirk]. Most of the species have been introduced from mycological series papers (e.g., Fungal Diversity Notes, Fungal Plant descriptions sheets, Fungal Systematic and Evolution (FUSE), New and Interesting Fungi, Fungal Planet description sheets) and been confirmed by DNA-based phylogenetic analyses, and ex-type cultures have been deposited in reputed culture collections. In this review, we discuss the importance of fungal cultures in the taxonomy of ascomycetous taxa (both lichenized and non-lichenized but except yeasts), culture collections as centers for preserving DNA and their limitations. Moreover, the importance of environmental sequencing and other culture-independent approaches to overcome the limitations of culture-dependent techniques in future mycology and its use in discovering missing species are also discussed.

## 2. Cultures in Traditional Taxonomy (i.e., Morphology-Based Taxonomy)

Morphological characters or phenotypic characters have played a vital role in mycology, since they have been used as the basic criterion for species recognition. Hence, the morphological species concept was the most popular species concept in traditional mycology, and a large number of species have been introduced based solely on morphological characters [7,8,9]. In addition to taxonomy, higher-level classification was also based on morphological characters. For example, Sutton [8] proposed a classification of coelomycetes based on the characters of conidiomata and the mode of conidiogenesis. However, the importance of cultures in artificial media has also been broadly discussed (e.g., Sutton [8] and Nag Raj [9]). Using cultures in early fungal studies was not very common, but Brown [10] used artificial media to observe the growth rate of fungi in atmospheres of different composition. In this section, we discuss the different aspects of cultures in the taxonomy of taxa in Ascomycota.

### 2.1. To Resolve Species Boundaries of Cryptic Species

Even though culture techniques were not popular at the time of publishing, Nobles [11] used culture characters, along with other macroscopic and microscopic characters, to identify wood-decomposing species of fungi. Subsequent publications that focused on taxonomy further discussed the isolation methods, choice of media, sporulation, and slide culture techniques [8,9,12,13]. Moreover, taxonomists have used culture characters to distinguish cryptic species in speciose genera, since morphological characters of conidia, conidiogenesis, and conidiomata are inadequate to establish species boundaries. For example, Boerema and Howeler [14], Boerema et al. [15,16,17,18,19,20], Boerema and Dorenbosch [21] and Boerema [22] used culture characteristics to distinguish several species within the genus *Phoma*. Additionally, Sutton [12] and Simmonds [23] used culture techniques (including appressoria production in slide cultures) to distinguish species of *Colletotrichum*.

Several studies have shown that it is important to observe the ‘anomalies’ of a particular taxon (mainly asexual) in a synthetic medium [9,24]. This is because different morphological structures are sometimes produced on media in culture that may be different from their appearance on the natural host [9]. Subsequent studies by Okada and Tubaki [25] and Holdenrieder and Kowalski [26] also supported this by reporting anomalies using culture-based techniques. To overcome these anomalies, Nag Raj [9] suggested using sterilized plant host material (preferably the leaves of the host from which the fungus was collected), which is placed in 2% sterilized water agar or plain agar, to cultivate the fungus.

### 2.2. To Reveal Pleomorphism and Propose a Single Name

Some taxa occur as both sexual and asexual morphs throughout their life history, but the two morphs are separated in time and space, making them pleomorphic taxa [27]. The links between sexual–asexual morphs and asexual–asexual morphs (synasexual morphs) have been established based on cooccurrence of both morphs on the same substrate [28], as well as by culture-based methods (in most cases, the development of asexual morphs in the cultures were observed from single ascospore isolates). Culturing a spore (i.e., single spore isolation of an ascospore, basidiospore or conidium) can help to reveal a new morph that has not previously been observed on the original host [29], and linking the morphs using cultures is much more reliable than simply using cooccurrence of both morphs on same host. Currently, a large number of studies have reported asexual morphs from single ascospore isolates, and in some cases both morphs have been new to science (e.g., Wanasinghe et al. [30] introduced *Haniomyces* J.C. Xu and reported a coelomycetous asexual morph from the culture), while in other cases asexual morphs have been reported for the first time from older sexual species (e.g., Li et al. [31] reported a coelomycetous asexual morph from cultures of *Cryptovalsa ampelina* [Nitschke] Fuckel).

Production of more than one asexual morph from a taxon (i.e., synanamorphs or synasexual morphs) has also been reported. These links have also been established based on hyphal connections (e.g., Hughes [32] for *Metacapnodium monillform* with *Capnobotrys Capnophialophora* and *Capnosporium* synanamorphs) or by using culture-based methods. For example, dichomera-like asexual morphs in cultures of *Botryosphaeria* and *Neofusicoccum* (as *Fusicoccum*) were reported by Barber et al. [33]. In a second example, *Cibiessia* was introduced by Crous et al. [34] with a *Readeriella* synasexual morph in vivo. Later, Crous et al. [35] proposed to adopt *Readeriella* over *Cibiessia*, as this link was well established based on the cultures. Some species of *Diaporthe* (=*Phomopsis*) produce different types of conidia, which are known as alpha, beta or gamma [36]. Mihaescu et al. [37] reported alpha and beta conidia from the pycnidia formed in cultures of *Phomopsis juglandina* (Sacc.) Hohn. (current name: *Diaporthe juglandina* (Fuckel) Nitschke). Therefore, it is important to maintain cultures of such genera to report the whole fungus or its holomorph.

The dual nomenclature of pleomorphic fungi has been abandoned. As such, proposing one name for two morphs is an important topic in current mycology [38]. Several studies have been carried out to propose one name for two morphs (i.e., adopt one name while suppressing the other name). Culture-based links have also been used to confirm the links between pleomorphic taxa and thus to arrive at one name (e.g., Wijayawardene et al. [39] proposed to adopt *Botryohypoxylon* Samuels and J.D. Rogers over *Iledon* Samuels and J.D. Rogers. *Iledon*, the asexual morph of *Botryohypoxylon*, was observed in the culture by Samuels and Rogers [40]).

### 2.3. To Isolate Mycobionts of Lichens

Culture-based studies of lichens, mainly mycobionts, have been thoroughly examined in traditional taxonomy. Ahmadjian [41] introduced a spore culture method to isolate mycobionts from lichen ascospores. Later, the same method was further developed by Yamamoto et al. [42] and Yoshimura et al. [43]. As spore culture methods showed some practical issues, including those related to spore discharge and spore germination, Yamamoto et al. [44] introduced a new method to isolate mycobionts using lichen thallus fragmentation called the “lichen tissue culture method” or the “Yamamoto method.” Lichens with a smooth surface or that which contains antibiotics show lower contamination rates in the Yamamoto method. However, lichens with soredia and soralia, as well as terricolous lichens, show a higher contamination rate [45].

## 3. Cultures to Extract Secondary Metabolites

Chemicals have been successfully extracted from cultured non-lichenized fungi and lichens and the former tested against many pathogens. Antibacterial, antifungal, superoxide scavenging, and tyrosinase inhibitory activity were tested by Yamamoto et al. [46] in cultured lichens of the families Cladoniaceae, Graphidaceae, Parmeliaceae, Umbilicariaceae, and Usneaceae. According to Kinoshita et al. [47], the cultured mycobiont of 26 species of lichens showed monoamine oxidase inhibition activity. To isolate the key genes involved in many activities, including secondary compound synthesis, symbiosis, and drought resistance in lichens, their cultures play an important role. Although cell aggregates that originated from natural lichen thallus fragments were composed of algal and fungal cells, their morphological differentiation is lacking since the algae and fungi act as the symbionts of the lichen [48].

A lichen is a nutritionally specialized phenotypic organism, and the interrelationship of the two different components is not properly understood. Therefore, culturing of lichens in vitro is far behind, although advances in experimental lichenology have been increased [48]. Some studies have reported monoamine oxidase inhibition activities of cultured mycobionts of selected species of lichens [47]. According to Behera et al. [49], the lichen *Bulbothrix setschwannsis* has successfully been cultured in vitro and it was possible to extract secondary metabolites. Furthermore, antioxidant, antityrosinase, and antibacterial properties of tissue cultured lichens were screened by Behera et al. [50]. Another study carried out by Koko et al. [51] investigated fungal-algal interactions specific to the lichen symbiosis using cultured *Usnea hakonensis* as a model system.

Many phenols such as depsides, depsidones, dibenzofurans, and pulvinates have been isolated from the fungal part of lichen cultures. Apart from that, many novel chemical constituents have been isolated from mycobiont cultures in the absence of the photobiont partner under stress conditions. Some examples include graphislactones from *Graphis scripta* var. *Pulverulenta* [52], graphisquinone from *Graphis desquamescens* [53], cristazarins from *Cladonia cristatella* [54], isofuranonaphthoquinones from *Arthonia cinnabarina* [55], a zearalenone derivative from *Baeomyces placophyllus*) [56], isoquinolines from *Amygdalaria panaeola* [57] and sesquiterpene derivatives from *Diorygma* sp. [58]. Overall, cultures are vital in both taxonomic and chemical studies (including chemotaxonomy) and subsequently for nomenclatural changes such as proposing one name.

## 4. Cultures as a Source for DNA

White et al. [59] pioneered the PCR and DNA sequencing techniques for fungal phylogenetics and ecology. Numerous culture-based studies and culture-independent studies have followed White et al. [59]. Indeed, cultures are widely used as the source for DNA, in addition to their application in conventional mycology for species identification and classification based on both morphology and phylogeny. Moreover, cultures have been used in chemotaxonomy for both lichenized and non-lichenized taxa [5,60,61].

### 4.1. Isolation

In nature, fungi can occur as a mixture of species, which makes it necessary to obtain DNA from a culture that is derived from a single species for DNA-based studies. In this respect, the isolation technique is a vital step towards obtaining a pure culture. However, these techniques depend on the habitat or host, life modes (endophytic, saprobes or pathogens), and their occurrence. The different isolation techniques used to obtain pure cultures have been comprehensively discussed by Senanayake et al. [62].

### 4.2. Culture Maintenance and Culture Collections

Maintaining pure cultures (a culture generated for a single species) prior to extraction of DNA is essential, since cultures can be contaminated by other air borne species (due to poor storage conditions or sterilization techniques) or destroyed by mites [62]. Using contaminated strains in DNA extraction can result in a mixture of DNA and erroneous phylogenetic analyses. The obtained cultures are stored and maintained in accessible culture collections, which thus serve as a resource for future studies [63,64,65,66]. Moreover, these cultures can be used as types when they are stored in metabolically inactive stages as in the International Code of Nomenclature for algae, fungi, and plants [67].

*Art. 8.4.* Type specimens of names of taxa must be preserved permanently and may not be living organisms or cultures. Nevertheless, cultures of algae and fungi, if preserved in a metabolically inactive state (e.g., by lyophilization or deepfreezing to remain alive in that inactive state) are acceptable as types).

*Art. 40.8.* For the name of a new species or infraspecific taxon published on or after 1 January 2019 of which the type is a culture, the protologue must include a statement that the culture is preserved in a metabolically inactive state.

The cultures obtained from a type is referred as the ex-type (ex typo), ex-holotype (ex holotypo), and ex-isotype (ex isotypo), and can be maintained as living cultures in accessible culture collections. Senanayake et al. [62] provided a comprehensive background of and recommendations for designating ex-type cultures. Moreover, dried cultures can be deposited as specimens in a fungarium [62]. These dried cultures can further enable us to observe if a culture produces its alternative morph or the same morph in culture.

## 5. Limitations of Culture-Based Studies

The field of fungal taxonomy has seen great progress in the past three decades, thanks to taxonomists who have widely used DNA sequences in species identification and classification. As an example, Abd-Elsalam et al. [68] pointed out the importance of maintaining culture collections of phytopathogens. Moreover, linking sexual–asexual morphs or synasexual morphs [35,39], resolving species complexes [69], delimiting generic boundaries of polyphyletic taxa [39], and epitypification of old species [70] have also been targeted in recent DNA-based taxonomic studies. A large number of species have been introduced based on DNA sequences and the respective cultures are deposited at accessible culture collections. Sequences generated from these cultures are deposited at GenBank for future research purposes.

Nevertheless, several taxonomic studies have reported the presence of a large number of unculturable fungi along with other culturable (mostly endophytic) fungi. Tejesvi et al. [71] carried out a study to reveal both culturable and unculturable taxa in *Rhododendron tomentosum* and obtained ITS and 5.8S rDNA sequences from 11 unculturable and 18 culturable species. Interestingly, the sequences generated from uncultured taxa by Tejesvi et al. [71] did not show high similarities with those obtained from culturable taxa of *Rhododendron tomentosum* and those from unculturable taxa are available in GenBank. Hence, Tejesvi et al. [71] emphasized the necessity of studying of both culturable and unculturable taxa in taxonomic studies and other studies such as screening for secondary metabolites. Nevertheless, numerous studies have been carried out to study endophytic taxa in the past three decades, and these studies have revealed new culturable taxa [72].

In addition to unculturable and non-sporulating endophytic fungi, a large number of environmental fungi (which are sporulating in nature) can be recognized as ‘unculturable’ taxa on synthetic media (e.g., *Rubroshiraia bambusae fide* Dai et al. [6]). These taxa can be highly host specific and/or sensitive to environmental parameters (such as salinity and pH). Thus, providing natural conditions may be helpful for growing them in culture (e.g., Nag Raj [9] suggested the use of sterilized plant leaves along with water agar).

Crahay et al. [73] stated that storing the cultures of ectomycorrhizal (ECM) fungi is important for ‘maintaining their genetic, phenotypic, and physiological stability’, but this could apply to other groups of fungi as well. Long term maintenance of cultures can result in the loss of viability or sporulation ability [74,75], pathogenicity, or virulence of pathogenic species [68,76,77,78]. Moreover, Crahay et al. [73] reported a reduction in the capability of fungal strains to form ECM associations, as well as in the effectiveness of fungi to enhance plant health, after being subcultured several times. These impacts could be due to ‘continuous subculturing which can lead to mutations and selection pressure on the organism’ [73]. These issues raise doubts about the efficiency and feasibility of maintaining cultures of ECM fungi, pathogens, and parasitic fungi, in addition to highlighting the need for improved methods to retain both viability, morphological features, and genetic stability (without mutations) over long periods of time [68].

In addition to type specimens (e.g., holotype, isotype or paratype), taxonomists currently deposit ex-type cultures (which are generated from type specimens) in culture collections. It is highly recommended to deposit ex-type cultures of novel taxa in accessible culture collections (Art. 8.4, 40.8) and some studies have encouraged mycologists to maintain a personal culture collection (e.g., the culture collection [CPC] of Pedro W. Crous). In most of the well-known culture collections, maintaining the strains on agar slants, storage in mineral oils or demineralized, sterile water, lyophilization, storage at −70 °C, and cryopreservation are the most common techniques [62,73]. Nevertheless, these methods are easy to contaminate by other microorganisms or mites (e.g., agar slants) in addition to being ‘labourintensive and spaceconsuming’ [73].

Currently, we lack DNA sequences for a large number of known taxa that have been described based only on morphological characters. According to Forin et al. [79], only 35,000 species are known from their DNA sequences. Several recent studies have been successful in the epitypification of old taxa as well as obtaining ex-epitypes. Two examples are [39], who designated the epitype of *Camarosporium propinquum* (Sacc.) Sacc. as *Pseudocamarosporium propinquum* (Sacc.) Wijayaw.), and Crous et al. [80], who designated the epitype and ex-epitype strain of *Arthrinium caricicola* Kunze. Taxonomists are encouraged to use the sequences generated from ex-type (or ex-isotype or ex-paratype or ex-neotype) cultures in their studies, including species identification and phylogenetic analyses. However, it may be a challenge to recollect the older taxa and epitypify them, due to land-use change at the site where a particular taxon was first collected (i.e., change from forest to agricultural) or due to the poor condition of the original descriptions or materials (holotype or isotypes) with which to compare new collections. Hence, old material would be the only option to obtain DNA, but it is difficult to obtain cultures from these old taxa since the viability of spores is lower. Thus, direct extracting DNA from old material has been suggested in some studies [79,81,82].

Fungi can be observed on the substrate (which was collected from the environment, i.e., based on morphological character or morphological species concept [83]) or after induced sporulation on plant surfaces; however, diversity can be overlooked when the fungi do not sporulate [84]. Several studies, e.g., [75,85,86,87,88] have shown that a large number of species exist as mycelial (vegetative) propagules that never produce spores (or undergo sporulation) or are unculturable.

Using the criterion of ‘inheritable character discontinuities’ [89] for delimiting species is a widely accepted method, and the phylogenetic species concept in association with DNA-based species identification is the most popular approach among taxonomists [83,89]. Moreover, DNA-based identification is also in agreement with a consolidated species concept (i.e., combining the morphological, ecological, and phylogenetic species concepts) *fide* [90]. Apparently, the consolidated species concept is in agreement with the ‘polyphasic species concept and integrative taxonomy’ [91]. Hence, to determine the ‘species’ and subsequently to predict the species number, DNA sequences are essential.

Nevertheless, a large number of studies will still depend on cultures to obtain DNA. Apparently, this is due to the conventional or traditional practices that rely on phenotypic or morphological characters. When describing a new species, it is widely accepted to provide a morphological description and DNA sequences (barcode) data to assist in species identification [92]. Aime et al. [29] stated that ‘the key element of a description is the demonstration that a specimen or culture represents a species that is distinct from previously described species’. However, Hongsanan et al. [93] mentioned that traditional molecular methods are appropriate only for those species that can be cultured (on synthetic media) and for fast growing species. We agree with this statement since fast growing, culturable taxa can inhibit or depress or overgrow slow growing species.

Wang et al. [94] mentioned that ‘the majority of the extant fungal diversity produces no distinguishing morphological structures that are visible or describable’. These taxa have been referred to as ‘‘dark taxa’’ or ‘‘dark matter” [95,96,97,98,99,100,101]. Seifert [96] emphasized the importance of GenBank records of ‘uncultured fungus’, ‘uncultured soil fungus’, and ‘uncultured endophytic fungus’ to catalogue all fungi. Hence, novel technologies are increasingly useful to reveal unculturable, dark fungi, while we attempt to determine their desired culture conditions. Figure 1 summarizes the basic steps of both culture-dependent and culture-independent methods.

Sanger sequencing [102,103] was the most popular sequencing technique until the mid-2000s and can only sequence specimens individually [104]. Margulies et al. [105] and Shendure et al. [106] developed ‘sequencing-by-synthesis technology’ to overcome the major drawbacks of Sanger sequencing. This new revolutionary technology, high-throughput sequencing (HTS) or ‘next-generation sequencing’ (NGS) [107,108], currently plays a vital role in microbial and mycological research.

## 6. Culture-Independent Methods

For many years, morphological and other phenotypic characteristics have been, almost exclusively, used to study fungal diversity and to classify fungi taxonomically [109]. Recent advances in DNA and RNA sequencing technologies, first from isolated individual organisms and later on from complex environmental samples, allowed us to rapidly gain knowledge on fungal diversity, ecology, taxonomy and related scientific fields, along with investigating fungal communities in an integrative way. Environmental genomics, also named metagenomics (a term was introduced in Handelsman et al. [110]), are thus applied to the study, at different levels, of complex fungal communities sampled directly from the environment. Its main advantages are that there is no need to perform previous culturing or isolation of the fungi inhabiting the target samples as well as the undesirable DNA cloning steps are avoided. These methods are based on the use of innovative platforms, available to the scientific community since 2004, which offer high speed and low-cost massive sequencing services. Depending on the aim of the study, sequencing of DNA and/or RNA-specific (PCR-amplified) or untargeted (shotgun approach as per Eisen [111]) genetic regions may provide large amounts of information. To cover these data representing all genetic material from the fungal component of the microbiota, the term mycobiome has been introduced by Ghannoum et al. [112]. Numerous pipelines such as PIPITS [113], CloVR-ITS [114], PipeCraft [115], and FindFungi [116] have been developed to filter and screen these sequence data linked to the mycobiome (see Appendix A for different methods that has been used in molecular taxonomy).

When using DNA, valuable information on microbial diversity and genomes contained in the particular environmental sample under study is produced, shedding light on the fungal taxonomic diversity occurring within it. In addition, RNA-sequencing can help to uncover active taxa and functions such as metabolic activities and potential functional roles associated with those fungi previously detected through DNA sequences [117]. Integrating both DNA and RNA data can provide a richer interpretation of fungal communities, leading to being able to address questions linked to ecosystem functioning. The number of metagenome projects investigating the presence and activity of fungal communities in environmental samples has been rapidly increasing [118]. These studies of the mycobiome have been focused on samples from diverse sources such as water (e.g., Brumfield et al. [119]), air (e.g., Calderón-Ezquerro et al. [120], soil/sediments (e.g., Samson et al. [121]), plants (e.g., Nguyen et al. [122], Yang et al. [123]) and humans (e.g., Hall et al. [124], Hanger et al. [125]). The number of complete fungal genomes and fungal environmental DNA/RNA sequences in public databases has grown considerably, reaching more than a billion reads just considering the ITS region [126]. Nevertheless, in comparison to prokaryotic organisms, the fungal component of natural communities remains understudied [127].

The advent of high-throughput sequencing methods over the past decade has revolutionized ecological and evolutionary studies of microbes, including fungi, by enabling sequencing analysis of a large number of species at an efficient cost and speed. Most of evolutionary and ecological studies of fungi thus far have focused on a few fragments of a genome, particularly ribosomal RNA genes. These methods have enabled us to identify many unknown fungi at the level of order and class, with implications to fill gaps in the fungal tree of life [128]. Rapid development in the technologies used in sequencing longer gene fragments has great potential to increase the accuracy of the method and identification of new clades [129]. There are also an increasing number of complete or nearly complete genomic studies, which shed unprecedented light on the evolutionary history of fungi [82,130,131]. As fungal genomes are relatively small (generally 30–100 Mb), it is possible to obtain a near-complete genome using culture-independent methods, as has been done using the metagenomics approach on *Inocybe* fruiting bodies [132]. These genomes can not only help to resolve the phylogeny and elucidate the evolution of nutritional modes in fungi, they also can inform us about their metabolic pathways and potentially provide information on their media requirements that can facilitate culturing them. Finally, this method allows us to sequence and identify old specimens [82].

The investment (or cost) of HTS methods is comparatively lower than in Sanger sequencing but throughput of former is higher than in the latter [133,134,135]. Moreover, HTS methods are less time consuming to process and this is regarded as an advantage [134,135]. In a broad picture, per base cost has become less than in other, more conventional technologies [136]. As a result, the number of metabarcoding projects has increased exponentially [137] (e.g., Global Soil Mycobiome and Funleaf, FunaAqua, Funhome projects).

As a general conclusion to these investigations, different abiotic factors such as soil pH, temperature or organic matter content have been suggested as drivers for shaping fungal communities in the environment [131,132,138]. It also seems clear that different microbial fractions (prokaryotes and eukaryotes) should be targeted together, if possible, since these groups are known to be closely linked as a result of cooccurring and interacting at different levels [118,131,132]. Below, we discuss the most important uses of CIM in modern ascomycetous studies.

### 6.1. Dark Fungi and Culture-Independent Techniques to Detect Them

Fungi that are not visible but exist only from genetic materials are known as dark fungi [98]. However, the term ‘fungi’ has “traditionally” been used to refer to taxa that are distinguished based on their morphological characters and other physiological characters [139]. The traditional DNA-sequencing or Sanger sequencing method [102] that have been used to sequence specimens individually are suboptimal for processing complex environmental samples from large-scale studies [104]. Nevertheless, the dark fungi are efficiently uncovered based on their genetic material (environmental DNA) by using metagenomics techniques as the main method for identification of dark taxa [129]. Simply put, metagenomics provides ‘a culture-independent genome analysis of entire microbial communities of a particular environmental niche’ [140]. These metagenomics studies reveal a large diversity of undescribed fungi, thus encouraging us to integrate fungal studies with ecological studies [131,132,138]. The data generated from high-throughput sequencing would also be important for revealing novel taxa which can be utilized in industry [140], in clinical aspects [141,142,143,144], in studies of ecosystem health [145,146], and studies of fossil fungi [147].

It is widely accepted that HTS is efficient in the detection of a broad range of taxa (which could be known taxa or novel taxa) in a relatively short time and at a low cost [2]. Different HTS methods are being used to detect unrevealing taxa in a broad range of environments, including soil and sea water [100,118,133,148,149,150,151]. Some species or even higher taxa have been observed infrequently or rarely cultivated [152]. Schadt et al. [153] predicted the possibilities of the existence of novel fungal lineages in soil. Tedersoo et al. [138] regarded that approximately ‘80% of all soilinhabiting fungal taxa cannot be identified at the species level, and 20% cannot be reliably assigned to known orders’.

Indeed, several new clades have been uncovered for fungi using HTS methods. Discovering *Archaeorhizomycetes* fungi from soil is one of the important studies since the members of this class are rarely reported with cultures [154,155]. Tedersoo et al. [128,129] showed that these uncultured fungi (which represent higher-level taxa) revealed by HTS are scattered throughout the fungal tree of life. Jones et al. [156,157] introduced the new phylum *Cryptomycota* (now *Rozellomycota*) from diverse environments (including soils, marine and freshwater sediments, freshwater planktonic samples, and oxygen-depleted environments [158]). De Beer et al. [159] introduced the genus *Hawksworthiomyces* with four new species based on cultures and one species based on environmental sequences. Moreover, De Beer et al. [159] emphasized the importance of using environmental sequences along with cultures and morphological data in naming hidden taxa. Nevertheless, the species names that are based only on sequences (as the type) have not been accepted in the International Code of Nomenclature for algae, fungi, and plants [67] (see below the section on ‘towards sequence-based nomenclature’). Taken together, HTS appears to be a reliable strategy for uncovering hidden taxa [126,154]. Below, we discuss the use of HTS in detection of endophytic taxa and marine inhabiting dark taxa.

#### 6.1.1. Endophytic Taxa; an Example of a Life Mode That Needs More Work

Quantitative and qualitative research on the community structure of plant fungal endophytes via culture-dependent techniques alone can pose a number of limitations. Endophytes are mutually beneficial endosymbionts that colonize internal living plant tissues throughout their lives or during part of their life cycle without causing any apparent disease symptoms [160,161]. Traditionally, surface-sterilized plant tissues are inoculated onto artificial growth media in order to isolate fungal endophytes [162,163]. However, these techniques can only retrieve culturable fungal endophytes, whereas unculturable endophytes will not be revealed, often providing inaccurate estimates of endophytic diversity. Moreover, isolation of culturable taxa can frequently depend on surface sterilization techniques, characteristics of the growth media used for isolation and incubation conditions such as time, temperature, pH, and interspecific competition amongst the endophytes [164]. Certain endophytic fungi may require growing conditions hitherto unknown. Moreover, with culture-dependent techniques, there is a likelihood of fast-growing fungal isolates inhibiting/masking slow-growing isolates on isolation media, often rendering them invisible [165,166]. The final endophyte community assembly of a host plant is a delicate balance between the host, environment, and the fungi that compete with one other. Therefore, it could be predicted that more adaptable and fast-growing fungi are abundant in endophytic communities (although this is yet to be entirely proven), thus increasing the isolation frequency of such fungi [167]. As a result of all these factors, there is a high chance of slow-growing isolates not being accurately represented in an endophyte community when derived via a solely culture-dependent approach. Long incubation periods, use of a higher number of small plant tissue segments or macerations, use of different culture media, and removal of fast-growing isolates from isolation media might increase the chances of extracting slow-growing endophytes. Nevertheless, latent and quiescent fungal endophytes might still be difficult to retrieve using these methods.

Recent fungal endophyte isolation attempts via both culture-dependent and culture-independent techniques from the same host plant have revealed largely different outcomes in terms of species richness, composition and diversity, whereas culture-independent techniques (e.g., high-throughput sequencing) have often yielded greater numbers. For example, endophyte identification studies in *Pinus taeda*, *Vitis vinifera* and *Dysphania ambrosioides* via HTS have resulted in greater species richness when compared to culture-dependent methods, and have revealed distinct clades. However, the same studies have further revealed the absence of certain easily culturable taxa in the endophytic community structure derived via culture-independent techniques (e.g., HTS) [168,169,170]. Fungal endophyte communities elucidated by HTS likely depend more on the DNA extraction method, primers, and the platform used for sequencing and analysis. Therefore, it is important to always combine culture-dependent techniques with culture-independent DNA-based techniques to obtain an accurate and complete estimation of the fungal endophyte diversity and composition within host plants.

#### 6.1.2. Marine Fungi as an Example

More studies could be focused on different aquatic environments, especially those associated with marine ecosystems where fungal biodiversity has yet to be heavily explored [171]. Most marine fungal taxa have been studied using culture-based methods. Bubnova et al. [172] developed an isolation method for samples of sea littoral and sub-littoral sediments. Overy et al. [173] reviewed different isolation techniques, such as direct isolation from sea foam, direct plating, particle filtration and dilution to extinction plating, damp chambers, baiting stations and in situ culturing for marine fungi. Furthermore, Raghukumar [174] extensively reviewed appropriate methodological tools to study various aspects of marine fungi, including techniques on culturing (direct detection and culturing, plating, baiting, and culturing). In addition to the culture-based research, Singh et al. [175], Li et al. [176] and Zhang et al. [177] studied sediments from marine environments using DNA metabarcoding (environmental sequencing) and confirmed a higher fungal diversity. We predict that a higher fungal diversity occurs in the shallow waters of the continental shelf which receive nutrient rich waters from upwellings and coastal currents. Therefore, areas of the continental shelf are rich in both habitat diversity and overall biodiversity [178]. As such, it is recommended that research be expanded to explore the fungi in these areas, as well as their ecological roles in shallow coastal ecosystems such as mangrove forests, seaweed meadows, sea grass beds, coral reefs, and mud flats using metagenomics. Moreover, greater attention has been directed towards fungal diversity in extreme deep-sea environments [179,180,181] and ocean microbiome [182] sea sediments [176], and both culture-dependent and culture-independent methods have revealed the presence of novel fungal phylotypes in these areas, including new taxonomic groups [179].

### 6.2. Studies on Older Herbarium Specimens

Hawksworth and Lücking [2], Hawksworth and Rossman [183] and Brock et al. [184] considered ‘existing reference collections’ to be important in revealing missing species of fungi. Precise and systemic observations of older herbarium materials are potentially useful in identifying new taxa. For example, it has been predicted that 70,000 species remain to be described and more than half of these already have been pre-served in existing collections [185]. Bruns et al. [186] discussed the possibility of using old specimens to extract DNA (ancient DNA). However, due to degradation, fragmentation, and the deamination, the DNA quality and quantity can be low, reducing the efficacy of PCR [187]. Forin et al. [79,81] used specimens of *Nectria* and *Peziza* from P. A. Saccardo’s mycological collection to extract DNA and revisit the taxonomic status of particular taxa. In both studies, Forin et al. [79,81] used high-throughput sequencing to overcome the DNA fragmentation and exogenous DNA contaminants. Daru et al. [188] is an interesting study which showed that old specimens harbored unrevealed endophytic species. Moreover, they experimentally proved that HTS is effective in discovering endophytic taxa from older specimens rather through the use of culturing. Smith et al. [187] used HTS for fungarium specimens of powdery mildews (*Erysiphales*) and found that it was possible to extract DNA from 25 years old specimens. A subsequent study by Smith et al. [189] used NGS to extract DNA from a powdery mildew (*Podosphaera* spp.) in the Victorian Plant Pathology Herbarium (VPRI). The fungus has been collected during the period of 1889 to 2008 on cherry and three other host plant genera from Australia. Results confirmed that all the species of powdery mildews belonged to *P. clandestine*, thus demonstrating the possibility of using HTS in pathology and plant protection. A recent study by Larkin et al. [190] used HTS for detection of fungi from formalin-fixed, paraffin-embedded tissues.

Taken together, HTS can be used to retrieve high-quality DNA from type specimens (and other reference specimens) deposited in fungaria, which are often more than 100 years old [82]. Since HTS can be used to analyze a mixture of species, old specimens with more than one taxon could be sequenced to reveal unknown species (e.g., sooty molds and species complexes).

## 7. Towards Sequence-Based Nomenclature

Since the introduction of HTS techniques [105,106], a broad range of high-throughput sequencing devices with different chemistries and detection techniques has been introduced commercially [104,191]. These have greatly facilitated a large number of studies in different disciplines (such as taxonomy, ecology, and paleomycology) over the past two decades. Such studies have generated a large amount of metagenomic DNA (mgDNA, mainly ITS sequence data) from different environmental samples (e.g., soil and water). Leinonen et al. [192] proposed a public repository, Sequence Read Archive (SRA) to store HTS data from eDNA. This database was established as a separate section of the International Nucleotide Sequence Database Collab-oration (INSDC) [192,193]. However, most of these environmental sequences data have not been linked to any specimens nor were any formal species names given (i.e., species name) [93,194].

Seifert [96] questioned the future perspectives of the sequences which have been deposited at GenBank as ‘uncultured fungus’, ‘uncultured soil fungus’, and ‘uncultured endophytic fungus’. However, currently, UNITE and FungalTraits, have taxonomically and functionally annotated taxa to a great proportion. Lücking and Hawksworth [126] regarded that ‘over 1 billion fungal ITS reads (1,222,062,203), with an average length of 375 bases’ are currently available in SRA and this number is higher than the number of fungal barcodes (sequences) in GenBank. Nevertheless, the number of precisely identified sequences in GenBank is likely much smaller, as some sequences can be chimeric, are not properly identified, or wrongly labelled or result from contaminated cultures [126]. Regarding the sequences (or Operational Taxonomic Units) generated from in environmental samples, most are not properly identified. Tedersoo et al. [128] reported that a large percentage of environmental sequences generated from soil samples belong to undescribed fungal taxa. Presumably, these sequences could represent old taxa which have not yet been sequenced. Huang [167] suggested that we should incorporate sequences generated from NGS and sequences from cultures in introducing novel taxa. However, the ICNafp does not allow us to recognize voucher-less taxa (dark taxa) as accepted species.

Art. 40.1. Publication on or after 1 January 1958 of the name of a new taxon at the rank of genus or below is valid only when the type of the name is indicated.

Nevertheless, depositing these sequences without any proper name will be problematic. Several studies have suggested assigning ‘informal names’ for these sequences [128,152]. Moreover, Hibbett et al. [152] proposed implementing a ‘candidate species category for *Fungi’* to reduce the communication errors due to informal names. Kõljalg et al. [195,196,197] introduced the Species Hypothesis concept for encoding and communicating described and undescribed taxa at various sequence similarity levels across research groups. Lücking et al. [91] and Lücking and Hawksworth [126] encouraged that sequence-based nomenclature would be the only solution to address the sequences generated from environmental sequencing while Hongsanan et al. [93] and Thines et al. [198] discussed its disadvantages. In addition, Lücking et al. [91], Hibbett et al. [152] further supported a different nomenclature system which is outside of ICNafp, but as a provisional name system similar to what is used in bacterial nomenclature. 

## 8. Future Directions

Discovering missing taxa in Kingdom *Fungi* is one of the most interesting research topics among taxonomists and ecologists today. Major fungal estimation studies [3,199,200,201,202] have emphasized the importance of further research on overlooked or less studied life modes and habitats (e.g., lichenicolous taxa, rock inhabiting taxa, and insect gut fungi), along with assessing cryptic species in order to discover novel taxa. Hyde et al. [203] discussed the importance of systematic, long-term studies of all types of fungi in relatively undisturbed forests. Endemic host species and widespread plant genera (such as species in the Rosaceae, *Eucalyptus* spp., and *Proteaceae* spp.) could be harboring more taxa than we might expect. Moreover, large numbers of taxa have yet to be discovered from less extensively studied life modes and less extensively studied geographical regions.

Nevertheless, the number of species introduced each year during the past decade is approximately 1200–2000. Publication series such as the Botanica Marina series, Fungal Diversity notes, Fungal Biodiversity Profiles, Fungal Systematics and Evolution—New and Interesting Fungi, Mycosphere Notes and Fungal Planet have contributed approximately 2000 novel taxa in the past decade, and are dedicated to the publication of novel taxa [203]. All these species are described based on morphology and/or DNA sequences analyses. Although assumptions are being made on where to find missing taxa, taxonomists have apparently restricted their studies to only a few life modes/habitats that have a major impact on humans, such as phytopathogens [200]. However, recent metabarcoding studies provide a broader picture of the unseen diversity of fungi in different ecological niches (however, taxonomists must also follow current rules when introducing novel taxa as in Code (i.e., the ICNafp), thus ‘ignoring’ the OTUs generated by metagenomics).

Among these metagenomic studies, soil is recognized as the ‘most diverse and densely populated microbial habitat on Earth, harboring high taxonomic and functional fungal diversity’ [134]. Metagenomics studies of soil in tropical rain forests would provide more novel taxa (i.e., OTUs). At the same time, other life modes such as litter-inhabiting taxa and nematode-trapping fungi should also be broadly studied.

## 9. Conclusions

Morphology-based species identification has been used extensively in traditional taxonomy. However, due to morphological plasticity and divergent evolution, taxonomists have recognized the limitations in this approach. DNA sequences have been used since 1990 [59] for species identification and fungal classification. Integrated taxonomy and the consolidated species concept, which uses both morphology and DNA sequence data, are now broadly accepted by modern taxonimists. In both traditional taxonomy and modern taxonomy, fungal cultures play vital roles. Culture characteristics, appressoria production, and asexual morph production in culture are important characters in identification and characterization. Taxonomists are encouraged to maintain living cultures of novel taxa as ex-type cultures and ex-epitype cultures of epitypes (of older species described only on the basis of morphology). These culture collections can be regarded as fungal genetic collections and can be used to extract DNA [200]. However, numerous species do not produce fruiting structures in nature or in cultures, and some taxa are unculturable. Hence, it is important for taxonomists to understand that morphology-based taxonomy and a dependence on cultures to obtain DNA is not always successful. At the same time, fungal estimation studies have indicated that these unculturable and non-sporulating taxa are a major component of the missing taxa yet to be described [2,200,201].

Currently, NGS or metagenomics techniques are playing important roles in environmental sequencing studies. Numerous sequences (OTUs or dark taxa) from unculturable taxa have been deposited in GenBank. Some of these sequences represent known species, while a large number of them are new to science and appear to represent novel lineages in Kingdom Fungi. Hence, several studies have emphasized the importance of metagenomics in discovering missing taxa in different life modes (e.g., Tennakoon et al. [204] discussed its important in discovering saprobic leaf litter fungi). However, currently, the ICNafp does not allow naming taxa without type material. Thus, several studies have discussed the necessity of proposing a nomenclature system for environmental sequences as a separate system not covered by the ICNafp [91,126,205]. We agree that there is a need for establishing a provisional nomenclature system, since it is the only way to answer the question, ‘where are the missing species?’ On the other hand, the taxa revealed from NGS or environmental sequences could be discovered in multiple locations. Since there is no proper nomenclatural system, one taxon could be given more than one name, and easily cause problems in the final cataloguing of species. However, most of the OTUs or dark taxa are inadequate or have erroneous sequences, and some studies have questioned the standards. As such, this methodology must be improved and become more accurate in order to accurately estimate the number of species [203].

## Figures and Tables

**Figure 1 jof-07-00703-f001:**
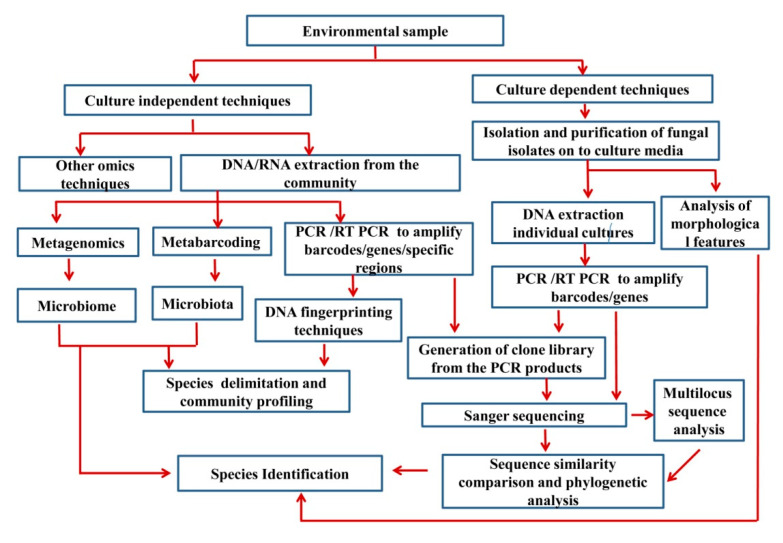
Basic steps in both culture-dependent and culture-independent methods.

## Data Availability

Not applicable.

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
