# Peer review of "Current Insight into Culture-Dependent and Culture-Independent Methods in Discovering Ascomycetous Taxa"

_jof, 2021, doi:10.3390/jof7090703_

Round 1

Reviewer 1 Report

This article deals with a topic that must be discussed in the field of fungal diversity.

However, there are some minor correnctions to be changed.

Please, check the detail from attached document.

Author Response

Dear Reviewer

Thank you for reviewing this MS. We have correct the MS based on your comments. Could you please see the responces in attachment. 

Reviewer 2 Report

This review paper tackles the issue of determining fungal diversity using culture dependent and culture independent methods. As such this is a huge topic especially when one considers the diversity of kinds of fungi and substrates on which they may occur. In general the authors seem to be referring to ascomycetes rather than basidiomycetes such as mushrooms including both saprobic and mycorrhizal fungi, jelly and polypore fungi and rusts and smut fungi. I suggest that this be acknowledged and perhaps even mentioned in the title of the article.

The progression of topics is logical and follows the chronology of the changes in fungal diversity studies. However, I suggest that the introductory paragraph is the place to include a summary of what Is currently known about fungal diversity i.e. how many species are estimated to occur, how many have been described, rather than placing them later in the article such as under section 8.

The article includes some repetition that I hope the authors will eliminate. Occasionally the authors repeat that many fungi are not culturable. This could be stated in the opening paragraphs and not repeated. Also, that fact that “morphologic characteristics have been used almost exclusively to classify fungi” is stated several times, although not true for yeast fungi.

I found this paper frustrating in that no topic is covered thoroughly and many of the paragraphs are filled with ideas that do not follow one from the other i.e. non-sequiturs. Perhaps if the review topic were limited to non-lichenized ascomycetes this would be less of a problem. There is so much to cover that none of it is done thoroughly. If the topic were limited, it might be possible to convey the excitement of NGS of one kind of substrate such as soil or decaying wood in which an incredible amount of fungal diversity has been discovered.

The problem of non-sequiturs is especially evident under section 2. For example, the second sentence of paragraph two concerns the “ observation of growth rates of fungi in atmospheres of different composition.” That topic doesn’t seem relevant to this review. The next sentence ln 78 is more in line with the topic of the review, although it seems that a word is missing after “is an outstanding taxonomic in which—taxonomic ?? in which? Then a sentence on general techniques, followed by a sentence on defining species on Phoma, and a sentence on Colletotrichum. The next paragraph is on asexual fungi both of which are represented in the previous paragraph.

Ln. 122, the correct name for Phomopsis juglandina is now Diaporthe juglandina (Fuckel) Nitschke

Covering the one name for fungi issue in one paragraph is difficult and probably should not be covered here except as a completed issue, i.e. not relevant to the use of cultural vs. culture independent methods.

The last four paragraphs of section 2 should be deleted. Instead the previous paragraphs should be expanded.

High throughput sequencing and next generation sequencing are the same. I suggest the authors stick to one of these labels.

Some of the headings include paragraph on topics that do not relate to the heading such as the paragraph on chemical extracted from lichens and their testing against pathogens. That does not seem to be a topic for this paper. One might consider these chemicals as non-morphological characteristics used in the identification of lichens but that is a huge topic in itself. I suggest that lichens not be considered in this paper.

Ln 252, in regard to personal culture collections, I strongly agree with citing them publications because for the most part these collections are neglected and discarded with the mycologist retires. It is much better for cultures to be deposited in permanent repositories in which they will be maintained in perpetuity for others to use. After all, that’s what makes any research science—the ability to be repeated by others.

I suggest that the authors refer to unculturable,unidentifiable fungi as dark fungi rather than dark matter fungi,

Section 8 could start with the third paragraph as this is an interesting topic about which much more can be said.

Author Response

(The authors gave the same response as above.)
